# Modified SCOPE (mSCOPE) Score as a Tool to Predict Mortality in COVID-19 Critically Ill Patients

**DOI:** 10.3390/jpm13040628

**Published:** 2023-04-02

**Authors:** Stavroula Zanelli, Agamemnon Bakakos, Zoi Sotiropoulou, Andriana I. Papaioannou, Evangelia Koukaki, Efstathia Potamianou, Anna Kyriakoudi, Evangelos Kaniaris, Petros Bakakos, Evangelos J. Giamarellos-Bourboulis, Antonia Koutsoukou, Nikoletta Rovina

**Affiliations:** 11st Respiratory Department, Medical School, National and Kapodistrian University of Athens, “Sotiria” Chest Hospital, 11527 Athens, Greece; 24th Department of Internal Medicine, Medical School, National and Kapodistrian University of Athens, “Attikon” Hospital, 12462 Athens, Greece

**Keywords:** COVID-19, mortality, mSCOPE score, critically ill patients, ICU

## Abstract

Introduction: Efficient clinical scores predicting the outcome of severe COVID-19 pneumonia may play a pivotal role in patients’ management. The aim of this study was to assess the modified Severe COvid Prediction Estimate score (mSCOPE) index as a predictor of mortality in patients admitted to the ICU due to severe COVID-19 pneumonia. Materials and methods: In this retrospective observational study, 268 critically ill COVID-19 patients were included. Demographic and laboratory characteristics, comorbidities, disease severity, and outcome were retrieved from the electronical medical files. The mSCOPE was also calculated. Results: An amount of 70 (26.1%) of patients died in the ICU. These patients had higher mSCOPE score compared to patients who survived (*p* < 0.001). mSCOPE correlated to disease severity (*p* < 0.001) and to the number and severity of comorbidities (*p* < 0.001). Furthermore, mSCOPE significantly correlated with days on mechanical ventilation (*p* < 0.001) and days of ICU stay (*p* = 0.003). mSCOPE was found to be an independent predictor of mortality (HR:1.219, 95% CI: 1.010–1.471, *p* = 0.039), with a value ≥ 6 predicting poor outcome with a sensitivity (95%CI) 88.6%, specificity 29.7%, a positive predictive value of 31.5%, and a negative predictive value of 87.7%. Conclusion: mSCOPE score could be proved useful in patients’ risk stratification, guiding clinical interventions in patients with severe COVID-19.

## 1. Introduction

COVID-19, which is caused by the novel coronavirus SARS-CoV-2, originally emerged in Wuhan, China in December 2019 as a cluster of patients with pneumonia [1]. On March 11 2020, COVID-19 was declared as a pandemic by the World Health Organization (WHO) and was characterized as a public health emergency of international concern [2].

The clinical presentation of COVID-19 seems to differ among patients, and it ranges from completely asymptomatic infection to severe respiratory failure caused by pneumonia, often requiring admission to the intensive care unit (ICU) and/or leading to death [3]. SARS-CoV-2 infection is usually followed by an incubation period, which varies between 2–14 days, after which symptoms usually appear. In terms of severe COVID-19, the mean period of time reported between the appearance of symptoms to death ranges from 6 to 41 days, depending on the patients’ age and the grade of immunocompetence [4]. Several studies reported that elderly (>60 years) patients represented the population group with the highest disease rates and the most rapid progression compared to younger patients [5,6]. Since severe COVID-19 often progresses to severe disease, resulting in respiratory failure and ARDS, supportive care with the use of mechanical ventilation, still plays the most decisive role on the treatment of critically ill patients [7,8]. Disease severity has been shown to depend on numerous risk factors, such as demographic characteristics (age, sex ethnicity), comorbidities, the presence of organ dysfunction, or systemic inflammation, which most times is implicated by the elevation of specific biomarkers, such as plasma creatinine, troponin, C-reactive protein (CRP), ferritin, D-dimers, or hepatic failure indicators (AST/ALT) [9,10], or it could be related to the severity of respiratory failure [11]. It is proved that severe pneumonia due to COVID-19 is associated with systematic hyperinflammation, accompanied by several inflammatory mediators (cytokines and chemokines) [12]. The SAVE MORE trial was specifically focused on using a countable biomarker called suPAR (soluble urokinase plasminogen activator receptor) and reported that in patients with levels above the cutoff of 6 ng/mL, and early treatment with recombinant interleukin-1 receptor antagonist was associated with favorable outcomes [13,14,15]. Furthermore, during the randomized control phase 3, SAVE MORE trial, a composite score, named SCOPE (the Severe Covid Prediction Estimate), which included four different biomarkers of systemic inflammation, endothelial activation, and coagulopathy (CRP, D-dimers, IL-6 and ferritin), has been proved useful in predicting progression to respiratory failure or death among patients admitted to the hospital due to COVID-19 pneumonia [9,16]. However, on the basis that IL-6 is not measured in several hospital settings, making the calculation of the SCOPE score difficult, a modified SCOPE (mSCOPE), consisting of three different laboratory test variables (Ferritin, D-Dimers and CRP), was introduced to be more widely used. The mSCOPE score has been shown useful in predicting progression to respiratory failure or death among patients admitted to the hospital due to COVID-19 pneumonia. Although the reliability of the score is based on the accuracy of the measurements of the three included parameters and can be considered high when measurements are performed using standardized methodology in certified laboratories, the validity or the mSCOPE has been evaluated only in the main study [9], in which it was validated in two similar independent cohorts. In the sub-group analysis using the same concentration quartiles of CRP, D dimers, and ferritin, mSCOPE showed similar high negative predictive value with SCOPE score [9]. 

Although mSCOPE score seems to be a useful predictor of clinical outcome and mortality in hospitalized COVID-19 patients, to our knowledge, there are no data regarding its usefulness as a predictor among patients admitted to ICU. Accordingly, the aim of the present study was to validate an abridged SCOPE score as a predictor of mortality in critically ill patients with COVID 19 pneumonia who required ICU admission.

## 2. Materials and methods 

### 2.1. Study Design

This is a retrospective observational study conducted in the 1st Intensive Care Unit (ICU) department of the National and Kapodistrian University of Athens at “Sotiria Chest Diseases Hospital” in Athens, Greece. A total of 268 patients have been included in this study, all of whom were hospitalized at some point from September 2020 until January 2022 in the ICU department due to severe COVID-19. All data were retrieved from the electronical medical files of each patient. It should be noted that part of the patients’ data used in this research has already been used in another publication by Koukaki et al [17]. 

In all patients, demographical and laboratory characteristics, comorbidities, and disease severity, according to the APACHE score, were recorded. The SCOPE SCORE was calculated, and the outcome of the patient during his/her stay in the ICU was also recorded. The study was approved by the Ethics Committee of the hospital (Ref No 23464).

### 2.2. Calculation of the Severe COVID Prediction Estimate (SCOPE) Score

The Severe COVID Prediction Estimate (SCOPE) score uses four laboratory variables to express the patients’ inflammatory response. Originally, SCOPE index was comprised of four parameters, C-reactive protein (CRP), d-dimers, ferritin, and interleukin-6 (IL-6). Each of the four biomarkers is allocated a score ranging from 0–3 (Table 1), and a threshold of ≥6 has been associated with a high sensitivity (almost 90%) with progress to severe respiratory failure or death the next 14 days and a high negative predictive value, which ranges to 96.7% [9]. However, since in our cohort a regular measurement of IL-6 was not performed, we opted to use the modified SCOPE index (mSCOPE), using the three remaining parameters (CRP, D-dimers, and ferritin), which has also been found to be a predictor of outcome in COVID-19 patients [9].

### 2.3. Study Participants

An amount of 268 out of 317 consecutive patients admitted in the ICU for severe COVID-19, were included in the study. The flow chart of the study participants is presented in Figure 1. SARS-CoV-2 infection was confirmed with reverse transcription polymerase chain reaction (RT-PCR) test. All included patients had a fully available medical history. Patients with missing data regarding their medical history and/or laboratory data, as well as those who were transferred to our department after a prolonged stay in another ICU, were excluded. 

### 2.4. Charlson Comorbidity Index and APACHE Score Calculation

The Charlson Comorbidity Index (CCI) [18] and the APACHE II score [19] were measured within the first twenty-four hours from the patient’s admission to the ICU. As for CCI, we used data found in the medical records of each patient stored electronically in medico//s, manufactured by Siemens Medical Solutions, as well as electronical files of medical prescriptions for each patient, which include the specific ICD-10 diagnoses describing each condition. CCI evaluates a vast number of comorbidities, and higher scores are predictors of increased mortality [18].

The APACHE II score was calculated based on the vital signs of each patient on admission, as well as their laboratory tests during the same day, both recorded in their medical files [19]. All laboratory tests were performed in the Sotiria Chest Diseases Hospital Laboratory to ensure that no discrepancies would exist between values from different laboratories.

### 2.5. Study Endpoints

The primary endpoint was to assess whether the mSCOPE index (including CRP, D-dimers and Ferritin) could be used as a predictor of mortality in patients admitted to the ICU due to severe COVID-19. Secondary objectives included the association between mSCOPE index and maximum ventilatory requirements and the presence of significant comorbidities.

### 2.6. Statistical Analysis

The normality of distributions was checked with Kolmogorov-Smirnov test. Data are presented as *n* (%) for categorical variables, and they are presented as mean ± SD for normally distributed, and they are presented as median (interquartile ranges) for skewed numerical variables. Comparisons between groups were performed using chi-square tests for categorical data, as well as unpaired t-tests or Mann-Whitney U-tests for normally distributed or skewed numerical data, respectively. Correlations were performed with Spearman’s correlation coefficient. Overall survival time was calculated from admission to the ICU until death. Patients discharged alive from the hospital were censored at the date of exit. Kaplan-Meier estimates were used to describe and visualize the effect of categorical variables. 

For the analysis of the primary objective, survival analysis and Cox regression analysis were implemented. In detail, the times to death according to the presence of a characteristic or adverse event was evaluated with Kaplan-Meier survival curves and log-rank tests. Cox regression univariate and multivariate analyses were performed in order to evaluate the influence of each characteristic or score in ICU mortality. Significant confounders evaluated in Cox regression analyses included age, sex, APACHE II score, Charlson comorbidity index score, and mSCOPE score. Results are presented as hazard ratios (HR) with 95% confidence intervals (CI).

For the assessment of the performance of SCOPE score as a predictor of mortality in ICU, and receiver operating characteristics (ROC) curves were created by plotting sensitivity against 1-specificity. The area under the ROC curve (AUC) with 95% confidence intervals (CI) and its difference from 0.5 were calculated. Additionally, sensitivities, specificities, positive (PPV), and negative (NPV) predictive values were calculated for specific cut-off points. 

In order to evaluate the target sample size, a power analysis was performed using the Gpower software. Using an effect size of 0.5, as well as an allocation ratio of 0.25, a sample size of 210 patients was required to achieve a power of 90% using an alpha significance level of 0.05 (two-sided).

The statistical analysis was performed using the SPSS 23 statistical package (SPSS Inc., Chicago, IL, USA), and graphs were created using GraphPad Prism 5 (GraphPad Software Inc., La Jolla, CA, USA) and MedCalc 9 (MedCalc Software, Mariakerke, Belgium).

## 3. Results

### 3.1. Study participants

An amount of 268 patients, 189 (70.5%) males, admitted to the ICU for severe COVID-19, were included in the analysis. The median (IQR) age of the study subjects was 61.0 (51.0, 69.7) years. An amount of 33 patients (12.3%) were current smokers, and 78 (29.1%) were ex-smokers. Demographic and medical characteristics of the study subjects are presented on Table 2.

### 3.2. Correlations of mSCOPE with Disease Severity and Outcomes

mSCOPE correlated to disease severity according to the APACHE score (*p* < 0.001, r = 0.767) and to the number and severity of comorbidities according to the Charlson Comorbitidy index (CCI) (*p* < 0.001, r = 0.616). Furthermore, mSCOPE significantly correlated with days on mechanical ventilation (*p* < 0.001, r = 0.766), days of ICU stay (*p* = 0.003, r = 0.681), and days of survival (*p* = 0.001, r = −0.504).

### 3.3. mSCOPE Score and the Presence of Comorbidities

Although there was a significant correlation between the mSCOPE score and the Charlson Comorbidity index, there was no difference on mSCOPE score between patients who suffered from frequent comorbid diseases and those who did not. The results are presented in Table 3.

### 3.4. mSCOPE According to the Maximal Ventilation Requirements

An amount of 146 (54.5%) of patients were intubated, and 52 (19.4%) patients were using Venturi masks, and 61 (22.8%) patients were using non-invasive mechanical ventilation, and nine (3.4%) patients required ECMO. Patients who required mechanical ventilation and those required ECMO had higher mSCOPE scores compared to those who had lower ventilatory requirements, i.e., 7 (6, 9) vs. 6 (5, 7) vs. 7 (4.5, 8.0) vs. 8 (6, 9) for patients been intubated, using Venturi masks and using non-invasive mechanical ventilation and patients on ECMO, respectively, *p* < 0.001 (Figure 2). 

### 3.5. mSCOPE and Mortality 

An amount of 70 (26.1%) of patients died in the ICU. Patients who died had higher mSCOPE score compared to patients who survived [8 (6, 9) vs. 7 (5, 8), *p* < 0.001, respectively]. (Figure 3). 

ROC analysis was performed to evaluate the predictive value of mSCOPE on the patients’ outcome. In ROC analysis, mSCOPE was associated with the patients’ outcome, since ROC curve differed significantly from 0.5 (Figure 4, Table 4), and AUC = 0.643 (0.582, 0.701). A mSCOPE value ≥ 6 was a predictor of poor outcome with a sensitivity (95%CI) 88.57 (78.7, 94.9), specificity was 29.69 (23.3, 36.7), positive predictive value was 31.5 (25.1, 38.5), and negative predictive value was 87.7 (77.2, 94.5). 

Patients with a mSCOPE score ≥ 6 had a worse survival compared to patients with lower scores (*p* = 0.004, Log Rank text). The Kaplan-Meier curve comparing patients with mSCOPE ≥ 6 and < 6 is shown on Figure 5.

In Cox regression analysis, independent predictors of mortality were the patients’ APACHE score on admission, the CCI, and the mSCOPE score (Table 5). 

Finally, in our study cohort, 42 patients (15.7%) had pneumo-mediastinum or pneumothorax during their stay in the ICU. These patients were 17/198 (8.6%) among survivors and 20/70 (28.6%) among non survivors, and *p* < 0.001 (chi-square test). Interestingly, mSCOPE did not differ between patients who developed pneumo-mediastinum or pneumothorax and those who did not. 

## 4. Discussion

In this retrospective study, we have shown that mSCOPE correlated to disease severity on admission to the ICU and to the patients’ total health status according to the number of comorbidities. Furthermore, our results show that mSCOPE significantly correlated with days on mechanical ventilation, the length of stay in the ICU, and to the patients’ outcome. We have also shown that SCOPE was higher in patients with higher oxygen requirements, with patients requiring intubation and mechanical ventilation or ECMO to have higher scores compared to those adequately oxygenated with Venturi masks and/or non-invasive mechanical ventilation. Finally, in our study, mSCOPE was an independent predictor of mortality, and a score over 4 could be used to predict the patients’ outcome with a good sensitivity and negative predictive value. 

Previous studies have shown that, in some patients, SARS-CoV-2 infection activates innate and adaptive immune responses that induce an hyperinflammatory state associated with cytokine storm and severe viral pneumonia [20,21,22]. It has also been shown that elevated clinical inflammatory markers are prognostic of disease severity and mortality [21], while the prevalence of ARDS is higher in patients with elevated inflammatory markers [22,23]. Clinical outcomes are found to be related to dysregulated antiviral immunity and enhanced and persistent systemic inflammation [21,22], as highlighted by a decreased expression of interferons I/III and hyperinflammatory responses involving the release of proinflammatory cytokines (e.g., IL-6, IL-1, TNF, IL-8, and MCP-1) that are sustained over time. Consequently, antiviral and anti-inflammatory therapies have been implemented in the pursuit of timely and effective therapies. Dexamethasone has gained a major role in the therapeutic algorithm of patients with COVID-19 pneumonia requiring supplemental oxygen or on mechanical ventilation. Its wide anti-inflammatory action seems to form the basis for its beneficial action, taming the overwhelming “cytokine storm”. Beyond doubt, the RECOVERY trial showed a clear benefit of dexamethasone in critically ill patients with COVID-19 on mechanical ventilation at the time of randomization as compared to the usual care (28-day mortality of 29.3% vs. 41.4%) [24]. 

Therefore, there is justification for our initiative to seek for an association between mSCOPE and mortality in critically ill patients with COVID 19 pneumonia who required ICU admission. The main tool used in this study was the modified SCOPE score. The mSCOPE score consists of three different laboratory test variables (ferritin, D-dimers, and CRP) and has been shown useful in predicting progression to respiratory failure or death among patients admitted to the hospital due to COVID-19 pneumonia. Although the reliability of the score is based on the accuracy of the measurements of the three included parameters and can be considered high when measurements are performed using standardized methodology in certified laboratories, the validity or the mSCOPE has been evaluated only in the main study [9], in which it was validated in two similar independent cohorts. One major strength of the SCOPE score is its high negative predictive value, however, it may be argued that IL-6 is not measured in several hospital settings, making the calculation of the SCOPE score difficult. In the sub-group analysis using the same concentration quartiles of CRP, D dimers and ferritin mSCOPE showed similar high negative predictive value [9]. C-reactive protein, D-dimers, and ferritin, which are used for the calculation of mSCOPE in the present study, are biomarkers that are commonly found to be elevated during inflammatory processes [25]. Accordingly, our observation that mSCOPE score was higher in patients with severe pneumonia and higher oxygen requirements is possibly related to the fact that, in these patients, the inflammatory process was more prominent, leading to a more severe disease. 

APACHE II score is used to classify disease severity and predict mortality in critically ill patients [26], with higher scores being associated with more severe disease and a higher risk of death. However, a previous cohort study showed that APACHE II underestimated disease severity and mortality risk in COVID-19 patients [27]. In the present study, we have shown that both APACHE II score and mSCOPE score were independent predictors of mortality. However, compared to APACHE II, mSCOPE score is calculated more easily, since it only requires three widely available and non-expensive biomarkers, which are routinely measured in patients with severe COVID-19 admitted to the ICU. In this regard, mSCOPE represents a quicker and easier method to evaluate prognosis compared to the more complex APACHE II score. 

Our results show a significant correlation between Charlson Comorbidity Index (CCI) and mSCOPE score. CCI is known to be able to predict the risk of death within one year of hospitalization and is a simple index to evaluate patients’ prognosis [18]. It has been shown that individuals with chronic diseases present with overexpression of angiotensin-converting enzyme (ACE)-2 receptors that may increase susceptibility and severity of COVID-19 pneumonia [28]. Accordingly, patients with chronic comorbidities might be at a higher risk of developing a hyperinflammatory state, as can be observed by elevated CRP, ferritin, and d-dimers in their laboratory tests results and can possibly explain the observed correlation between CCI and mSCOPE score. Thus, the combination of comorbidities scoring systems and laboratory tests in the setting of severe COVID-19 pneumonia may be useful in predicting disease progression and outcome. However, the fact that mSCOPE scores did not differ between patients with and without specific comorbidities is probably related to the fact that the CCI describes a global impairment of the health status and seems to be more accurate compared to the presence or absence of one specific chronic disease. 

The association between mortality risk and elevated inflammatory biomarkers in patients with severe COVID-19 pneumonia was supported by several studies [29,30,31], while elevated serum CRP, PCT, D-dimer, and serum ferritin levels were associated with an increased composite poor outcome that included mortality, ARDS, and the need for ICU admission in patients with COVID-19 [32]. Similarly, another study has shown that, during the hyperinflammation stage of COVID-19, pneumonia, which is known to be associated with poor prognosis, as well as inflammatory biomarkers, such as CRP, d-dimer, and ferritin, are significantly elevated [13]. The aforementioned observations are in accordance with the results of our study, in which we have shown that patients who died in the ICU had a higher mSCOPE score compared to those who survived. 

There is evidence that patients in mechanical ventilation due to severe COVID-19 have increased risk of developing barotrauma and that these patients have poorer clinical outcomes, while barotrauma per se seems to be an independent predictor of in hospital mortality in severe COVID-19 [33,34,35]. Pneumothorax has been identified in up to 20% of mechanically ventilated patients and to almost 30% of patients on ECMO [36,37,38].

In the same vein, in our study 42 patients (15.7%) developed pneumo-mediastinum or pneumothorax during their stay in ICU, and, in accordance with previous studies, these events were related to poorer survival [33,38]. However, in our study, the mSCOPE score did not differ between patients who developed barotrauma and those who did not, and this observation leads to the hypothesis that this score provides a global evaluation of the acute inflammatory syndrome occurring in severe COVID-19 and does not represent alterations associated with lung mechanics. 

Our study has some limitations. First, this was a retrospective study based on the data available in the electronic records of our patients. However, by excluding patients with missing data in the parameters used for the calculation of mSCOPE, we have tried to optimize our cohort and achieve a high quality of data. Second, although the main validation of SCOPE in patients with COVID-19 used four parameters (CRP, ferritin, D-dimers, and IL-6), since in our department, a regular measurement of IL-6 was not performed, we opted to use the modified SCOPE index, using the three remaining parameters, which have been found to also be predictors of outcome in COVID-19 patients [9]. We have to admit that the sub-score consists of three biomarkers that are known to be elevated in patients with systematic inflammation, regardless of SARS-CoV2 infection. However, this simplified score was still able to differentiate patients with more severe disease and adverse outcome and had a high sensitivity on predicting death in our study cohort, although it is lacking the high specificity of the four parameter SCOPE score. Finally, we have excluded patients who were transferred to our department from another ICU, since these patients might be experiencing complications related to prolonged ICU stay and long-term mechanical ventilation, which might not be directly related to the severity of COVID-19.

In conclusion, despite the mass vaccination programs, SARS-COV-2 virus remains a great challenge for public health systems, as some of the patients still develop severe pneumonia and require admission to the ICU. Globally, the availability of ICU beds is limited and, hence, the development of accurate prognostic models in guiding clinical decision making and timely interventions is of vital importance. The biomarkers used in the mSCOPE score are inexpensive and widely available, and the score could be proved useful in the patients’ risk stratification and guiding clinical interventions in patients with severe COVID-19.

## Figures and Tables

**Figure 1 jpm-13-00628-f001:**
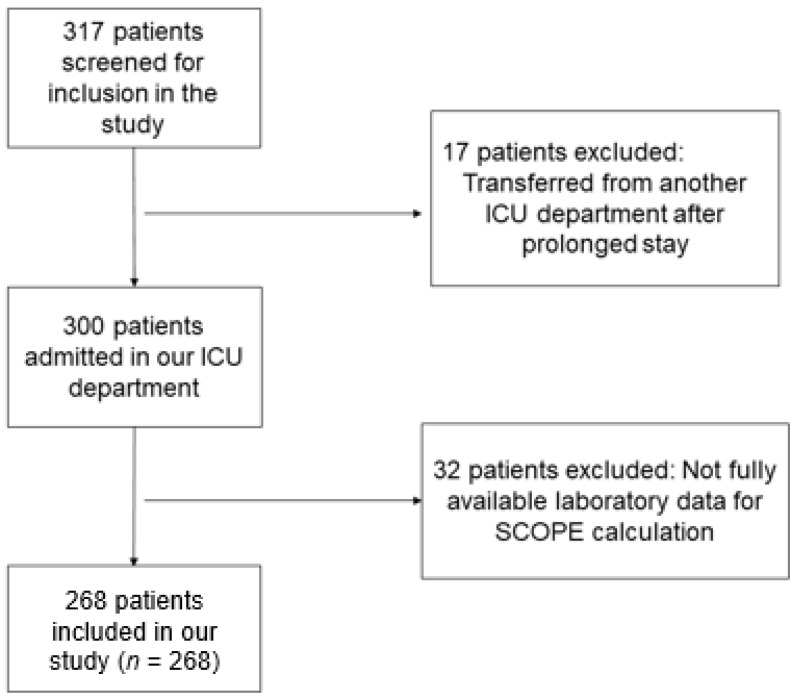
Flow chart of the study participants.

**Figure 2 jpm-13-00628-f002:**
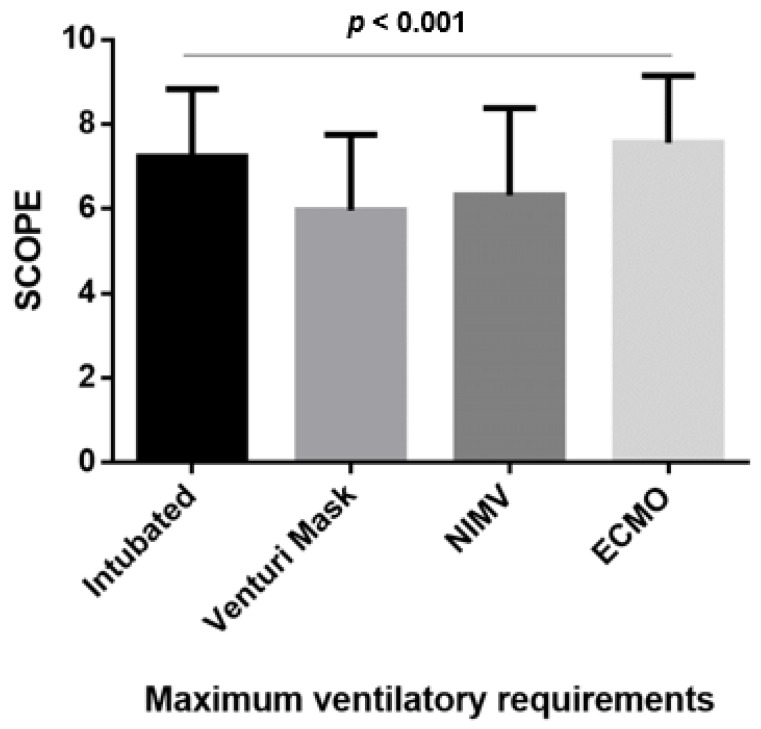
SCOPE according to respiratory requirements in the study subjects.

**Figure 3 jpm-13-00628-f003:**
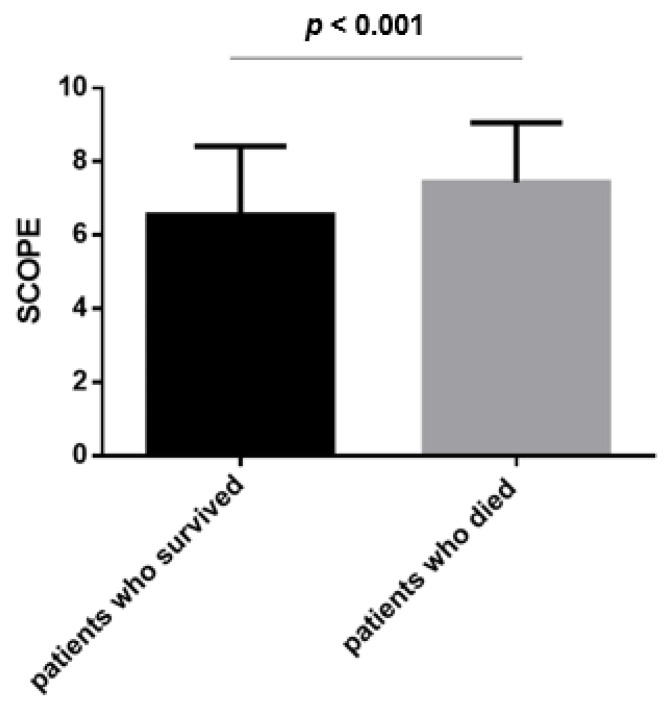
SCOPE according to the patient’s outcome.

**Figure 4 jpm-13-00628-f004:**
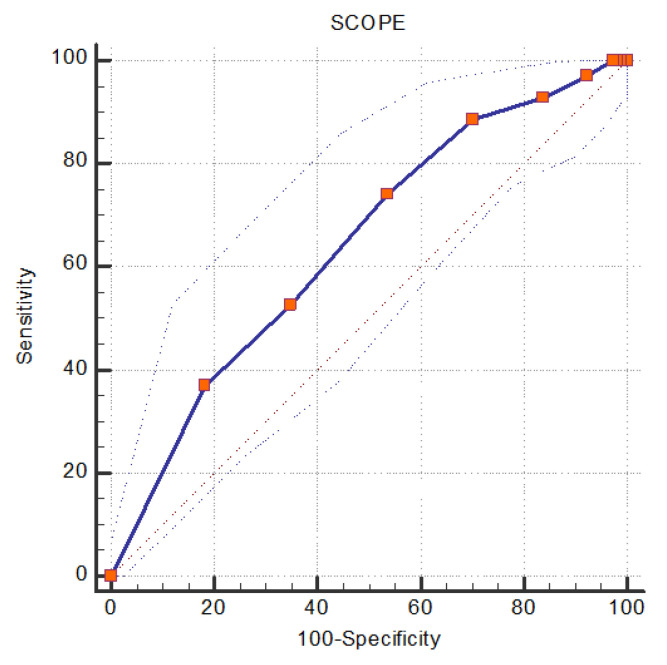
ROC Curve for the use of SCOPE as a predictor of mortality in patients hospitalized in the ICU for severe COVID-19.

**Figure 5 jpm-13-00628-f005:**
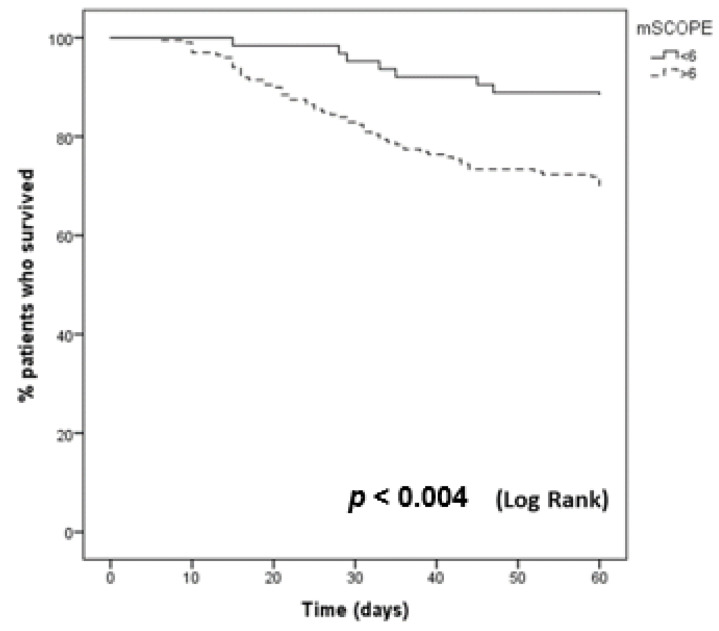
Kaplan-Meier curve comparing patients with mSCOPE ≥ 6 and < 6.

**Table 1 jpm-13-00628-t001:** Modified SCOPE index (three-parameter) calculation without IL-6.

D-Dimers (mg/mL)	CRP (mg/L)	Ferritin (ng/mL)	Points
0.10–0.40	0.3–25.0	10.0–225.0	0
0.41–0.57	25.1–45.0	225.1–450.0	1
0.58–0.90	45.1–85.0	450.1–750.0	2
>0.91	>85.1	>750.1	3

**Table 2 jpm-13-00628-t002:** Demographic and medical characteristics of the study subjects.

Variable	
Age (years)	61.0 (51.0, 69.7)
Sex (M) N (%)	189 (70.5)
BMI kg/m^2^	29.3 (26.0, 33.0)
Smoking (never/current/ex) N (%)	157/33/78 (58.6/12.3/29.1)
APACHE Score	11.0 (8.0, 14.0)
Comorbidities N (%) Respiratory disease Diabetes mellitus Arterial hypertension Thyroid disease Coronary disease	39 (14.6) 56 (20.9) 121 (45.1) 43 (16) 27 (10.1)
Ventilatory requirements N (%) Mechanical ventilation Venturi Mask NIMV ECMO	146 (54.4) 52 (19.4) 61 (22.8) 9 (3.4)
Length of stay in ICU (days)	10.0 (6.0,23.0)
Length of stay in the Hospital (days)	26.0 (17.0, 41.0)
Outcome (death) N (%)	70 (26.1)
mSCOPE	7.0 (6.0, 8.0)

Variables are presented as median (IQR) or as N(%) unless otherwise indicated. Abbreviations: BMI: body mass index, APACHE: Acute Physiology and Chronic Health Evaluation, NIMV: non-invasive mechanical ventilation, ECMO: extracorporeal membrane oxygenation, ICU: intensive care unit, mSCOPE: modified Severe COVID Prediction Estimate.

**Table 3 jpm-13-00628-t003:** Differences in mSCOPE scores among patients with and without significant comorbidities.

Comorbidity	mSCOPE		*p*-Value
	Patients w/o the Comorbidity	Patients with the Comorbidity	
Respiratory disease	7.0 (6.0, 8.0)	6.0 (5.0, 8.0)	0.176
Diabetes mellitus	7.0 (5.0, 8.0)	7.0 (6.0, 8.7)	0.202
Hypertension	7.0 (5.0, 8.0)	7.0 (6.0, 9.0)	0.306
Thyroid disease	7.0 (6.0, 8.0)	6.0 (5.0, 8.0)	0.101
Coronary disease	7.0 (5.5, 8.0)	7.0 (6.0, 9.0)	0.764

Abbreviations: mSCOPE: modified Severe COVID Prediction Estimate.

**Table 4 jpm-13-00628-t004:** ROC analysis for the predictive value of mSCOPE on ICU mortality in patients hospitalized for severe COVID-19.

	Optimal Cut-Off Point	Sensitivity (95% CI)	Specificity (95% CI)	PPV	NPV	AUC (95% CI)	*p*-Value
SCOPE score	≥6	88.57 (78.7, 94.9)	29.69(23.3, 36.7)	31.5 (25.1, 38.5)	87.7 (77.2, 94.5)	0.643 (0.582, 0.701)	<0.001

Abbreviations: PPV: positive predictive value, NPV: negative predictive value, AUC: area under the curve, CI: confidence interval, mSCOPE: modified Severe COVID-19 Prediction Estimate. In Cox regression analysis, mSCOPE score was an independent predictor of survival in our study cohort HR (95%CI) 1.219 (1.010–1.471), *p* = 0.039 (Table 5).

**Table 5 jpm-13-00628-t005:** Univariate and Multivariate Cox Regression analysis.

Variable	Univariate Analysis			Multivariate Analysis		
	HR	95% CI	*p*-Value	HR	95% CI	*p*-Value
Age	1.075	1.051–1.099	<0.001	0.984	0.952–1.017	0.332
Sex	0.882	0.521–1.494	0.640			
APACHE score	1.135	1.091–1.181	<0.001	1.090	1.038–1.145	0.001
CCI	1.672	1.501–1.864	<0.001	1.693	1.417–2.023	<0.001
mSCOPE score	1.297	1.121–1.500	<0.001	1.219	1.010–1.471	0.039

Abbreviations: CCI: Charlson Comorbidity Index, mSCOPE: modified Severe COVID Prediction Estimate, APACHE: Acute Physiology and Chronic Health Evaluation.

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
