# Peer review of "Modified SCOPE (mSCOPE) Score as a Tool to Predict Mortality in COVID-19 Critically Ill Patients"

_jpm, 2023, doi:10.3390/jpm13040628_

Round 1
Reviewer 1 Report
The aim of the study was to validate SCOPE score as a predictor of mortality in critically patients with COVID 19 pneumonia, who required ICU admission.
The study design is adequate and reproducible, the analysis is well conducted. The message is clear and could be relevant for the field of interest. The figures, tables and images are appropriate.
Author Response
Reviewer 1
The aim of the study was to validate SCOPE score as a predictor of mortality in critically patients with COVID 19 pneumonia, who required ICU admission.
The study design is adequate and reproducible, the analysis is well conducted. The message is clear and could be relevant for the field of interest. The figures, tables and images are appropriate.
Answer
We thank the reviewer for his/her comments.
Reviewer 2 Report
Dear authors,
Congratulations on this significant work.
Few comments:
- Kindly elaborate on sample size calculation, power analysis, effect size
- Kindly reflect on validity and reliability of the tools used. It is understandable that you are investigating predictive validity however, other forms of validity and reliability that have been established should be mentioned if possible.
Author Response
Reviewer 2
Congratulations on this significant work.
Few comments:
- Kindly elaborate on sample size calculation, power analysis, effect size
We thank the reviewer for his/her comment.
The effect size was calculated by taking the difference of mSCOPE between the two groups (i.e. survivors vs non survivors) and dividing it by the standard deviation of one of the groups. According to that the effect size was calculated in 0.5
Regarding the sample size calculation a power analysis was performed using the Gpower software. Using an effect size of 0.5, and an allocation ratio of 0.25 a sample size of 210 patients was required to achieve a power of 90% using an alpha significance level of 0.05 (two-sided).”A comment has been added in the manuscript as follows
“In order to evaluate the target sample size, a power analysis was performed using the Gpower software. Using an effect size of 0.5, and an allocation ratio of 0.25 a sample size of 210 patients was required to achieve a power of 90% using an alpha significance level of 0.05 (two-sided).”
-Kindly reflect on validity and reliability of the tools used. It is understandable that you are investigating predictive validity however, other forms of validity and reliability that have been established should be mentioned if possible.
Answer
We thank the reviewer for his/her comment. The main tool used in this study was the modified SCOPE score. The SCOPE score consists of 3 different laboratory test variables (Ferritin, D-Dimers and CRP) and has been shown useful in predicting progression to respiratory failure or death among patients admitted to the hospital due to COVID-19 pneumonia. Although the reliability of the score is based on the accuracy of the measurements of the three included parameters and can be considered high when measurements are performed using standardized methodology in certified laboratories, the validity or the mSCOPE has been evaluated only in the main study (PMC8872836) in which it was validated in two similar independent cohorts.
A comment has been added in the discussion section as follows:
“The main tool used in this study was the modified SCOPE score. The SCOPE score consists of 3 different laboratory test variables (Ferritin, D-Dimers and CRP) and has been shown useful in predicting progression to respiratory failure or death among patients admitted to the hospital due to COVID-19 pneumonia. Although the reliability of the score is based on the accuracy of the measurements of the three included parameters and can be considered high when measurements are performed using standardized methodology in certified laboratories, the validity or the mSCOPE has been evaluated only in the main study (PMC8872836) in which it was validated in two similar independent cohorts.”

Reviewer 3 Report
I have read your manuscript with interest. Although I appreciate the authors' effort, the manuscript needs major revision. The manuscript also benefits from extensive language editing.
Abstract
Please provide a brief SCOPE background in the introduction and also justify the study in terms of necessity and significance.
Introduction
If SCOPE predicted mortality and clinical outcome in hospitalized Covid-19 patients, it seems intuitive to me that it would do the same for critically ill patients. Instead of making the aim “to validate SCOPE score as a predictor of mortality in critically ill patients with COVID-19 pneumonia, who required ICU admission”, I would suggest you use “to validate an abridged SCOPE score as a predictor of mortality in critically ill patients with COVID 19 pneumonia, who required ICU admission”.
Paragraphs 3,4 and 5 should be rewritten to give better focus and sharpness to the presentation. Much of the information there is not germane to the background. In addition, SCOPE should be better reviewed in terms of its power and insufficiencies in order to properly situate this study, which is like an extension of the original SCOPE study
Materials and Methods
Were there only male patients? The use of “his” suggests that.
Aside from missing data and transfer from another ICU, were there other exclusion criteria? If not, why? Malignancy, primary immunodeficiency, renal failure, pregnancy, and prior anti-cytokine treatment etc., are all possible confounders.
Results
-In reporting your p values, be consistent with either the actual value of p or its acceptance level (use p<0.001 or p=0.003)
Discussion
Curiously, there was a significant correlation between the mSCOPE score and the Charlson Comorbidity index, but there was no difference in the mSCOPE score between patients based on comorbidities. Can you give possible reasons for this?
I think you could briefly discuss the performance of mSCOPE in the light of SCOPE since the former is an offshoot of the latter. How would SCOPE have performed here?
References should follow the same format.
Author Response
Reviewer 3
I have read your manuscript with interest. Although I appreciate the authors' effort, the manuscript needs major revision. The manuscript also benefits from extensive language editing.
We thank the reviewer for his/her comment. Language editing has been done according to reviewer’s suggestion.
Please provide a brief SCOPE background in the introduction and also justify the study in terms of necessity and significance.
We thank the reviewer for his/her comment. A short paragraph has been added in introduction and a more extensive argument has been added in discussion.
Introduction If SCOPE predicted mortality and clinical outcome in hospitalized Covid-19 patients, it seems intuitive to me that it would do the same for critically ill patients. Instead of making the aim “to validate SCOPE score as a predictor of mortality in critically ill patients with COVID-19 pneumonia, who required ICU admission”, I would suggest you use “to validate an abridged SCOPE score as a predictor of mortality in critically ill patients with COVID 19 pneumonia, who required ICU admission”.
We thank the reviewer for his/her comment. We modified the phrase according to his/her suggestion.
Paragraphs 3,4 and 5 should be rewritten to give better focus and sharpness to the presentation. Much of the information there is not germane to the background. In addition, SCOPE should be better reviewed in terms of its power and insufficiencies in order to properly situate this study, which is like an extension of the original SCOPE study
We thank the reviewer for his/her comment. We deleted completely paragraph 4, and paragraphs 4 and 5 are reformed according to his/her suggestion. We also added a paragraph describing the SCOPE and Mscope background so as to justify its use in this study.
Materials and Methods
Were there only male patients? The use of “his” suggests that.
As shown in Table 2 189 (70.5%) of patients were male. We modified “his” to his/her stay in the ICU so as to be clear.
Aside from missing data and transfer from another ICU, were there other exclusion criteria? If not, why? Malignancy, primary immunodeficiency, renal failure, pregnancy, and prior anti-cytokine treatment etc., are all possible confounders.
There were no other exclusion criteria. The study included pragmatic real-life cases of critically ill patients that usually are on multiorgan failure on their admission in the ICU.
Results -In reporting your p values, be consistent with either the actual value of p or its acceptance level (use p)
We modified the p values to be expressed as the actual p value, except for values <0.001

Reviewer 4 Report
The overall manuscript confirms informations already described in the literature.
I suggest to the authors to stratify the patients according to their vaccination status, considering also the time of the last vaccine dosein results and discussion. Indeed, there are some paper that suggest the importance of the timing of vaccination (PMID: 36507545 + 36301541 + 35608862) and the role of cellular mediated immunity. This would add some novelty.
In addition, an increasing volume of literature has associated the incidence of pnuemo-mediastinum or pneumothorax to worst outcome. Please report these data and discuss.
Author Response
Reviewer 4
The overall manuscript confirms information already described in the literature.
I suggest to the authors to stratify the patients according to their vaccination status, considering also the time of the last vaccine dose in results and discussion. Indeed, there are some paper that suggest the importance of the timing of vaccination (PMID: 36507545 + 36301541 + 35608862) and the role of cellular mediated immunity. This would add some novelty.
We thank the reviewer for his/her comment. The collected data refer to hospitalizations from September 2020 to January 2021. Most of that time vaccination was not available for the total of the population in our country (vaccines began to be available in March 2023 and for several months were available only for the elderly and for high risk patients). For that reason, the majority of our study population was unvaccinated or incompletely vaccinated which also seem to have contributed to the development of severe disease and requirements for ICU admission
In addition, an increasing volume of literature has associated the incidence of pneumo-mediastinum or pneumothorax to worst outcome. Please report these data and discuss.
Answer
We thank the reviewer for his/her comment.
In our study cohort 42 patients (15.7%) had pneumo-mediastinum or pneumothorax during their stay in ICU. These patients were 17/198 (8.6%) among survivors and 20/70 (28.6%) among non survivors, p<0.001 (chi-square test). Interestingly, mSCOPE did not differ between patients who developed pneumo-mediastinum or pneumothorax and those who did not.
A comment has been added in the results section as follows:
“In our study cohort 42 patients (15.7%) had pneumo-mediastinum or pneumothorax during their stay in ICU. These patients were 17/198 (8.6%) among survivors and 20/70 (28.6%) among non survivors, p<0.001 (chi-square test). Interestingly, mSCOPE did not differ between patients who developed pneumo-mediastinum or pneumothorax and those who did not.”
Furthermore, a comment has been also added in the Discussion section as follows:
“There is evidence that patients in mechanical ventilation due to severe COVID 19 have increased risk of developing barotrauma and that these patients have poorer clinical outcomes while barotrauma per se seems to be an independent predictor of in hospital mortality in severe COVID-19 [31-33]. Interestingly, in our study 42 patients (15.7%) developed pneumo-mediastinum or pneumothorax during their stay in ICU and in accordance to previous studies these events were related to poorer survival [31, 33]. However, in our study the mSCOPE score did not differ between patients who developed barotrauma and those who did not, and this observation leads to the hypothesis this score provides a global evaluation of the acute inflammatory syndrome occurring in severe COVID-19 and does not represent alterations associated with lung mechanics.”

Round 2
Reviewer 4 Report
No further comments